# A Multiplex Quantitative Polymerase Chain Reaction for the Rapid Differential Detection of Subgroups A, B, J, and K Avian Leukosis Viruses

**DOI:** 10.3390/v15091789

**Published:** 2023-08-23

**Authors:** Junfeng Dou, Zui Wang, Li Li, Qin Lu, Xinxin Jin, Xiaochun Ling, Zhengyu Cheng, Tengfei Zhang, Huabin Shao, Xinguo Zhai, Qingping Luo

**Affiliations:** 1Key Laboratory of Prevention and Control Agents for Animal Bacteriosis (Ministry of Agriculture and Rural Affairs), Institute of Animal Husbandry and Veterinary, Hubei Academy of Agricultural Sciences, Special One, Nanhuyaoyuan, Hongshan District, Wuhan 430064, China; djf0825@163.com (J.D.); wangzui@webmail.hzau.edu.cn (Z.W.); lili_0215@126.com (L.L.); luqin198909@126.com (Q.L.); jinxinxin@webmail.hzau.edu.cn (X.J.); cheery2221@163.com (X.L.); shhb1961@163.com (H.S.); 2Hubei Hongshan Laboratory, Wuhan 430064, China; 3Department of Animal Medicine, College of Life Science and Food Engineering, Hebei University of Engineering, Handan 056038, China; 4Department of Microbiology and Immunology, Dalhousie University, Halifax, NS B3H 4R2, Canada

**Keywords:** avian leukosis virus, multiplex quantitative PCR, subgroup A/B/J/K

## Abstract

Avian leukosis (AL), caused by avian leukosis virus (ALV), is a contagious tumor disease that results in significant economic losses for the poultry industry. Currently, ALV-A, B, J, and K subgroups are the most common in commercial poultry and cause possible coinfections. Therefore, close monitoring is necessary to avoid greater economic losses. In this study, a novel multiplex quantitative polymerase chain reaction (qPCR) assay was developed to detect ALV-A, ALV-B, ALV-J, and ALV-K with limits of detection of 40, 11, 13.7, and 96 copies/µL, respectively, and no cross-reactivity with other ALV subtypes and avian pathogens. We detected 852 cell cultures inoculated with clinical samples using this method, showing good consistency with conventional PCR and ELISA. The most prevalent ALV strain in Hubei Province, China, was still ALV-J (11.74%). Although single infections with ALV-A, ALV-B, and ALV-K were not found, coinfections with different subgroup strains were identified: 0.7% for ALV-A/J, 0.35% for ALV-B/J, 0.25% for ALV-J/K, and 0.12% for ALV-A/B/K and ALV-A/B/J. Therefore, our novel multiplex qPCR may be a useful tool for molecular epidemiology, clinical detection of ALV, and ALV eradication programs.

## 1. Introduction

Avian leukosis (AL) is a severe immunosuppressive disease of chickens caused by avian leukosis virus (ALV) and is a general term for various avian neoplastic diseases [1]. check all author names carefully AL has caused huge economic losses in the poultry industry and is one of the major diseases endangering China’s poultry industry [2]. In 2021, it was listed as one of the major avian diseases to be prevented in the National Animal Disease Surveillance and Epidemiological Survey Plan (2021–2025). ALVs are classified as members of the family *Retroviridae*, subfamily *Orthoretrovirinae*, and genus *Alpharetrovirus* [1]. Based on the viral envelope interference, host range, and cross-neutralization patterns, ALVs are divided into subgroups A–K [1]. Subgroups A–E, J, and K are specific to chickens, F and G are specific to pheasants [3], and H and I are found in partridges and quails, respectively [1]. Exogenous subgroups A, B, J, and K are the most common ALVs in commercial poultry in China, which can cause the host to produce different types of tumors. Although subgroup E is an endogenous virus with low or no pathogenicity, it is prevalent in the field and can mislead the diagnosis of exogenous viruses [4].

To prevent and control AL, several vaccines have been developed, but they do not provide sufficient protection for chickens [5,6]. Due to the absence of effective vaccines and drugs, rigorous eradication measures are the only effective way to eradicate ALV. Currently, commercial ELISAs are widely used to detect ALV antigens because they can analyze a large number of clinical samples simultaneously and rapidly. However, when detecting viruses in clinical samples, except for egg whites, this method cannot exclude the interference of endogenous viruses. More importantly, it is incapable of distinguishing different ALV subgroups. ALV-A/B/J/K not only represent the most prevalent strains but also pose the greatest threat to the poultry industry. Coinfections with ALV-J/K, ALV-A/J, and ALV-B/J have been observed in commercial laying hens. These instances of coinfection present a potential opportunity for recombination between distinct subgroups of ALV strains. Thus, a rapid and convenient method for molecular epidemiology and differential detection of ALV-A/B/J/K is essential. Molecular techniques such as multiplex polymerase chain reaction (PCR) and real-time PCR have been widely applied to identify the subgroups of ALV [7,8,9]. These methods can efficiently differentiate between two or three subtypes of ALV but cannot simultaneously distinguish ALV-A/B/J/K subtypes. In addition, ALV can generate genetic variability and evolve rapidly through RNA error-prone polymerase and frequent recombination events that lead to the emergence of many new mutant and recombinant strains [10,11]. Hence, it is necessary to improve the existing molecular detection techniques. In the present study, a multiplex real-time quantitative PCR (qPCR) was established to simultaneously detect four subtypes: ALV-A, ALV-B, ALV-J, and ALV-K.

## 2. Materials and Methods

### 2.1. Virus and Clinical Sample Treatment

ALV-A (XTM-21-01), ALV-B (SZ-19-01), ALV-J (SX-18-01), ALV-K (SH-20-01), avian influenza virus (H9N2, DY0602), Newcastle disease virus (NDV, TS09-C), infectious bronchitis virus (IBV, HB120), reticuloendotheliosis virus (REV, XY-18-01), fowl adenovirus serotype 4 (FadV4, HB1510), infectious laryngotracheitis virus (ILTV, WG strain), *Mycoplasma synoviae* (XY-1), and *Mycoplasma gallisepticum* (DY-1) were stored in our laboratory. Marek’s disease virus (MDV, CVI988 Rispens) vaccine was purchased from YEBIO (Qingdao, China). The *gp85* gene fragment of ALV-E (AY013303.1) was synthesized by Sangon Biotech (Shanghai, China). DF-1 cells were cultured in Dulbecco’s modified Eagle’s medium (DMEM, Thermo Fisher Scientific, Waltham, MA, USA) supplemented with 10% fetal bovine serum (FBS) and maintained in our laboratory. The cells were susceptible to exogenous ALV identified in chickens, while ALV-E was unable to proliferate.

A total of 852 clinical samples including tissues, plasma, egg white, and semen were collected from 5 different commercial poultry farms (Appendix A) in the Hubei provinces of China. After three rounds of freeze-thawing, the supernatants obtained from homogenized tissues and other samples were used for P27 antigen detection and DNA/RNA extraction. In addition, the treated samples were filtered through 0.22 μM membranes and then incubated with DF-1 cells for 2 h. After being washed three times with PBS, the cultures were maintained at 37 °C in a 5% CO_2_ atmosphere by adding DMEM containing 1% FBS. Meanwhile, uninfected DF1 cells were utilized as a negative control. After 7–9 days, the cultures were freeze-thawed three times, and the resulting supernatants were used for P27 antigen detection and DNA/RNA extraction.

### 2.2. DNA/RNA Extraction

Viral genomic DNA/RNA of the supernatant was extracted by a FastPure^®^ Viral DNA/RNA Mini Kit virus Kit (Vazyme Biotech, Nanjing, China). The cDNA was synthesized by the HiScript^®^ II Q RT SuperMix (Vazyme Biotech). The cDNA was preserved in an ultra-low temperature freezer at −80 °C.

### 2.3. Design of Primers and Probes

The *gp85* gene sequences of ALV-A/B/J/K/E were downloaded from the GenBank database. The GenBank entries used as reference sequences for region selection are presented in Table 1. Multiple sequence alignments were carried out to identify conserved regions of the ALV genome that exhibited inter-subgroup divergence using DNAMAN version 6.0.3.40 software (Lynnon Biosoft, San Ramon, CA, USA). Based on the sequence alignment results, three sets of primers and four sets of probes (Table 2) were designed using Primer Premier 5 software (Premier, Waterloo, ON, Canada). To prevent interference between fluorescence signals in this multiplex system, four fluorescent dyes with distinct wavelengths were chosen for the probes: Rhodamine X (ROX), Cyanine5 (CY5), Victoria (VIC), and FAM (Carboxy Fluorescein) were used to modify the probes for ALV-A, ALV-B, ALV-J, and ALV-K, respectively. All primers and probes were synthesized by Sangon Biotech.

### 2.4. Optimization of the Multiplex qPCR

The multiplex qPCR was performed with a 10 μL AceQ^®^ qPCR Probe Master Mix (Vazyme Biotech) combined with all primers, probes, templates, and nuclease-free water to a final volume of 20 μL. Primer and probe concentrations for ALV-A, ALV-B, ALV-J, and ALV-K were optimized to enable the simultaneous detection of all four viruses with minimum *Cq* values. In addition, the optimum reaction temperature was tested from 55 °C to 65 °C, with a temperature gradient of 1 °C. The optimal conditions were defined as those that allowed for the simultaneous detection of all four viruses with minimum *Cq* values. All qPCRs were conducted by QuantStudio^™^ Real-Time PCR (ABI, Natick, MA, USA).

### 2.5. Specificity of the Multiplex qPCR

The specificity of the multiplex RT-PCR was evaluated by testing the reactivity with the cDNA of ALV-A/B/J/K, H9N2, NDV, IBV, REV, ILTV, and the DNA of *M. synoviae* and *M. gallisepticum*, MDV, FadV4, DF-1, and ALV-E. Nuclease-free water served as a negative control.

### 2.6. Sensitivity of the Multiplex qPCR

To generate the plasmid standard, the detection targets of ALV-A/B/J/K were amplified and cloned into the pMD18-T vector, respectively. The concentrations of pMD-alvA, pMD-alvB, pMD-alvJ, and pMD-alvK were determined using a NanoDrop One/OneC spectrophotometer (Thermo Fisher Scientific). The plasmid standards concentrations for ALV-A/B/J/K were 4.0 × 10^6^, 1.1 × 10^6^, 1.37 × 10^6^, and 9.6 × 10^6^ copies/μL. Sensitivity and standard curve tests were conducted using a 10-fold dilution series of recombinant plasmids, ranging from 10^1^ to 10^6^ copies/µL as templates, and the results were compared to the PCR results. Nuclease-free water was used as a negative control. The limit of detection (LOD) was determined based on the lowest concentration of standard plasmids that consistently yielded a detectable *Cq* value. Additionally, the standard curves were generated by plotting the *Cq* values against the logarithm of standard copy numbers and calculating the R^2^ (correlation coefficient) values to assess linearity.

### 2.7. Repeatability of the Multiplex qPCR

The assay was performed three times with a 7-day interval, utilizing 10-fold dilutions of the positive standard plasmid ranging from 10^3^ copies/μL to the LOD, with three replicates per reaction. Each template comprised an equal concentration mixture of standard plasmids from four pathogens. The coefficient of variation (CV) for *Cq* values was calculated to evaluate repeatability across all concentrations.

### 2.8. Verification of the Multiplex qPCR

Mixed infections were frequently misdiagnosed due to the presence of pathogens at lower, undetectable concentrations. To mimic the occurrence of mixed infections, plasmid standards from two, three, or four pathogens were amalgamated at their LODs to serve as templates for the assay. The resulting simulated mixed infections were evaluated by multiplex qPCR.

Finally, the clinical samples and cell cultures inoculated with clinical samples were tested for P27 antigen using a commercial ELISA (IDEXX, Westbrook, ME, USA). According to the ELISA protocol, samples with S/P values greater than 0.2 are considered positive, while those with S/P values between 0.1 and 0.2 are classified as suspicious. Meanwhile, the extracted cDNAs of the clinical samples and cell cultures inoculated with clinical samples were screened for ALV-A/B/J/K using established multiplex qPCR and conventional PCR with the primers listed in Table 2. Multiplex qPCR results were compared to those from normal PCR and commercial ELISA.

## 3. Results

### 3.1. Establishment of a Novel Multiplex qPCR

Primers and probes were selected from the most divergent yet conserved region specific to each ALV subgroup based on *gp85*, ensuring specificity for detecting the target without cross-reactivity with other subgroups (Figure 1). Degenerate bases were inserted into probes to account for unavoidable point mutations when detecting ALV-A and ALV-B targets. The optimal qPCR system included a 10 μL qPCR Probe Master Mix, 0.3 μL primers (10 μM) for qALV-A/K, B, and J, 0.2 μL probes (10 μM) for ALV-A/B/J/K, 2 μL templates, and 5.4 μL nuclease-free water (Appendix A and Appendix A). The optimal reaction conditions for multiplex qPCR involved initial denaturation at 95 °C for 5 min followed by 40 cycles at 95 °C for 5 s and 60 °C for 30 s (Appendix A and Appendix A).

### 3.2. The Specificity of the Multiplex qPCR

To evaluate the specificity of the multiplex qPCR, the cDNA/DNA of major prevalent pathogens in chicken populations were detected as templates. Only ALV-A/B/J/K subgroup strains produced four positive fluorescence signals, and no positive signals were obtained with other viruses (H9N2, NDV, IBV, REV, ILTV, *M. synoviae*, *M. gallisepticum*, MDV, FadV4, ALV-E) and nuclease-free water (Figure 2).

### 3.3. Sensitivity of the Multiplex qPCR

To test the sensitivity of the multiplex qPCR, standard plasmids from four viral subtypes were mixed equally and diluted 10-fold with deionized distilled water. Dilutions ranging from 10^6^ to 10^1^ copies/µL were used for both multiplex real-time PCR and routine PCR detection under optimized conditions. The multiplex qPCR showed high sensitivity in detecting ALV-A, ALV-B, ALV-J, and ALV-K with LODs of 40, 11, 13.7, and 96 copies/µL, respectively, which were 10–100 times higher than conventional PCR methods (Figure 3). Four linear relationships were obtained between the copy number (*x*-axis) and quantification cycle (*Cq*) value (*y*-axis). The amplification efficiencies and correlation coefficients showed excellent linearity, with R^2^ values of 1.000 for ALV-A, 0.990 for ALV-B and ALV-J, and 0.998 for ALV-K (Figure 4). The *Cq* value demonstrated reliability within the range of 32–34 for LOD concentrations but exhibited unreliability beyond 35. Consequently, we established a positive cut-off threshold at a *Cq* value of 35.

### 3.4. Repeatability of the Multiplex qPCR

To assess the repeatability of qPCR, three different concentrations of standard plasmids were detected at a 7-day interval. The CV of intra- and inter-assays ranged from 0.27% to 0.72% and from 0.13% to 1.09% for the plasmid pMD-alvA, 0.55–0.83% and 0.43–0.95% for pMD-alvB, 0.17–0.29% and 0.25–0.82% for pMD-alvJ, and 0.77–0.85% and 0.33–0.58% for pMD-alvK, respectively. The CV of *C_q_* was <2%, indicating that the assay was highly reproducible (Table 3).

### 3.5. Evaluation of the Simulated Coinfections

To conduct a comprehensive coinfection simulation, all plasmid standards with concentrations at the LOD were utilized. This ensured that the results were accurate and reliable. The advantage of this approach lies in its ability to detect not only duplex or triplex coinfections but also quadruplex ones, enabling the detection of ALV-A/B/J/K, even at low concentrations within mixed infection samples (Figure 5 and Figure 6).

### 3.6. Evaluation Using the Clinical Samples

A total of 852 clinical samples, including tissues, plasma, egg whites, and semen, were detected using multiplex qPCR, conventional PCR, ELISA, and virus isolation. The positive rates of ALV for multiplex qPCR, conventional PCR, ELISA, and virus isolation were 15.96% (136/852), 15.14% (130/852), 18.90% (161/852), and 12.91% (110/852), respectively (Table 4). The ELISA method exhibited a higher positivity rate compared to other methods, especially when detecting albumen samples. Moreover, the positive rate of the multiplex qPCR was higher than conventional PCR and virus isolation. In the agreement assays, the multiplex qPCR demonstrated good concordance with the conventional PCR method (Appendix A). The poor consistency compared to ELSIA or virus isolation might be attributed to the potential interference of endogenous viruses or the presence of non-viable virions.

Furthermore, the cell cultures (n = 852) with clinical samples were determined by multiplex qPCR and conventional PCR (Table 5). The positive rates for ALV-A/J, ALV-B/J, ALV-K/J, ALV-A/B/K, and ALV-A/B/J were 0.70%, 0.35%, 0.23%, 0.12%, and 0.12%, respectively, as determined by multiplex qPCR, which was consistent with conventional PCR. Additionally, 100 samples (11.74%) were detected as positive for ALV-J alone by multiplex qPCR, while 97 samples (11.38%) were detected as positive by PCR. The three samples exhibiting disparate outcomes were identified with 0.1 ≤ S/P ≤ 0.2 utilizing ELISA, indicating that multiplex qPCR was more sensitive than both ELISA and conventional PCR. Single infections with ALV-A, ALV-B, and ALV-K were not identified.

## 4. Discussion

ALV is an avian oncogenic retrovirus that can cause growth retardation, reduced egg production, and immunosuppression [12]. Subclinical pathological and various neoplastic diseases have caused huge economic losses to the poultry industry [10]. Like all retrovirus diseases, AL is difficult to prevent and control with vaccines and drugs. At present, the only effective method is population purification. Commercial ELISA kits have been widely used, but they cannot be used for the identification of ALV subgroups, nor can they distinguish exogenous from endogenous ALVs, which is not conducive to our better understanding of the prevalence of ALV. Therefore, there is an urgent need to establish a simple, rapid, and low-cost detection method to improve eradication efficiency.

Currently, many molecular assays have been developed such as multiplex PCR, multiplex qPCR, loop-mediated isothermal amplification, and reverse transcription recombinase-aided amplification [7,8,13,14]. Due to the advantages of simple operation, short duration, and high sensitivity, multiplex qPCR is the most widely used in clinical diagnosis. Although double qPCR for detecting ALV-A/B and ALV-J/K has been reported, none of these assays can simultaneously detect the major circulating strains (ALV-A/B/J/K) in China. To achieve rapid detection of the four subgroups in ALV infection, a multiplex qPCR with high specificity and sensitivity was developed and optimized to detect ALV-A/B/J/K simultaneously.

Primer and probe design plays an important role in developing a successful multiplex qPCR. The classification of ALV subgroups is based mainly on the nucleotide sequence of the envelope protein (*gp85*) [15]. After comparing the sequences of ALV-A, ALV-B, ALV-J ALV-K, and ALV-E strains published in GenBank, we designed primers and probes for detecting ALV-A/B/J/K with the *gp85* gene. ALV-A/K shared a common pair of detection primers. The four probes were selected from a conservative region for each subgroup and did not show any complementary reaction with other ALV subgroups via BLAST search. However, some point mutations cannot be avoided when detecting the targets in ALV subtypes. To ensure the detection of more diverse field strains, the probes for detecting ALV-A/B were inserted with degenerate bases as needed. Additionally, four signals (FAM, ROX, CY5, and VIC) were chosen to detect and differentiate the four subgroups of strains based on their different wavelengths. These fluorescence signals could be detected simultaneously in the same reaction tube without interference.

The sensitivity and specificity of the multiplex qPCR were also critical. In the present study, the specificity of the multiplex qPCR was determined using different subgroup ALVs and various avian viruses. Only ALV-A/B/J/K strains produced four positive fluorescence signals, and no positive signals were obtained with other viruses and mycoplasmas (H9N2, NDV, IBV, REV, ILTV, *M. synoviae*, *M. gallisepticum*, MDV, FadV4, and ALV-E) and nuclease-free water. The LODs of the assay in the single-tube assay system were determined as 40 copies/µL for ALV-A, 11 copies/µL for ALV-B, 13.7 copies/µL for ALV-J, and 96 copies/µL for ALV-K, and they were about 10–100 times more sensitive than conventional PCR. The CVs of the assay were <2%, indicating good accuracy and reproducibility. Many benefits would be gained from the improved methods. Accurate detection of low levels of the virus is crucial for early diagnosis and prevention of ALV infection, especially when some strains (ALV-A and ALV-B) can cause severe subclinical symptoms despite minimal evidence of virus shedding [7]. The new detection method had a low LOD, which enabled more effective surveillance of the four ALV subgroups, benefiting population purification efforts. In the efficiency tests, 852 clinical samples were analyzed using the multiplex qPCR, conventional PCR, commercial ELISA, and virus isolation. The positive rate of the multiplex qPCR was higher than conventional PCR and virus isolation. However, the ELISA method demonstrated a higher rate of positivity compared to other methods, particularly in the detection of albumen samples. One potential explanation is that some of the egg whites exclusively contain viral components, while another reason could be attributed to the additional processing required for DNA extraction from egg whites. This is supported by the lower rate of virus isolation from the egg whites. For the other clinical samples, the higher positive rate of ELISA may potentially be attributed to the presence of ALV-E, which needs to be further investigated. Furthermore, when detecting cell cultures, the results of the qPCR were in almost perfect concordance with those obtained by conventional PCR and ELISA, except for three samples that exhibited 0.1 ≤ S/P ≤ 0.2 values using ELISA, suggesting that the multiplex qPCR was more sensitive than conventional PCR and ELISA. Therefore, results demonstrated that the novel multiplex qPCR exhibited high efficiency for the detection of ALV, with a remarkable level of specificity, and could be used to detect clinical samples, except egg whites.

Coinfection, which can be more pathogenic and cause greater economic losses, is now prevalent in the poultry industry [16,17]. Due to its severe immunosuppressive effects, ALV often coinfects with other viruses such as MDV, REV, and IBDV [18,19,20,21]. Mixed infections can also occur between different subgroups of ALV. Coinfection with ALV-A and ALV-J was found in the sarcomas of 817 broiler hybrids, while ALV-B and ALV-J were detected together in laying hens [7,22]. Mixed infections with ALV-J/K and ALV-A/J have also been reported in clinical samples [23,24]. In the present study, ALV-J remained the most prevalent strain in Hubei province, with coinfections with ALV-B/J (three cases), ALV-J/K (two cases), ALV-A/J (six cases), ALV-A/B/K (one case), and ALV-A/B/J (one case).

The majority of mixed infection cases were traced back to the same chicken farm, where the ALV-positive rate reached 37.3%. The likelihood of mixed infections was heightened, rendering the development of singular infections with other subgroups of viruses more challenging. Although the pathogenicity of coinfection between ALV subgroups is uncertain, it provides an opportunity for the recombination of different subgroups. In recent years, recombinant ALV strains have been identified with increasing frequency. The recombinant strains that have been reported include ALV-E/K, ALV-B/J, ALV-A/J, and ALV-J/K [24,25,26,27]. Additionally, an ALV-K, ALV-E, and ALV-J multiple recombinant strain was isolated from Chinese native chickens. Compared to the original strains, the recombinant strains may exhibit greater pathogenicity and epidemic potential [24,27]. Fortunately, recombination among four or more subtypes has not been detected. Given that coinfections with different ALV subgroups are common in the field and their high recombination frequency, developing a rapid clinical detection method is crucial for effectively controlling the spread of ALV infections.

In summary, we developed a multiplex real-time PCR that detected and differentiated the four common subtypes of avian leukosis virus (ALV-A, ALV-B, ALV-J, and ALV-K) with high sensitivity and specificity. This method was rapid, convenient, and practical for laboratory and clinical diagnoses, as well as useful in epidemiological studies and eradication programs.

## Figures and Tables

**Figure 1 viruses-15-01789-f001:**
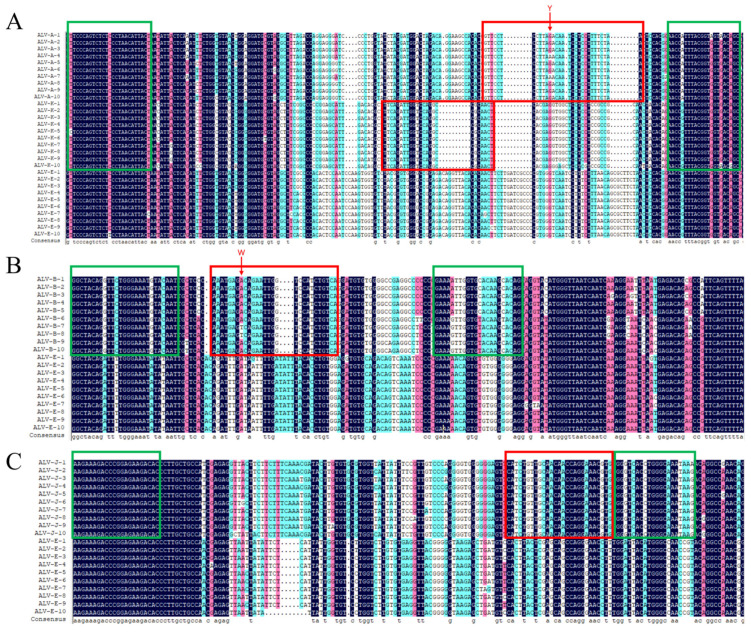
Design of probes and primers for detecting ALV–A, ALV–B, ALV–J, and ALV–K. The probes and primers of (**A**) ALV–A and ALV–K, (**B**) ALV–B, and (**C**) ALV–J were designed using Primer Premier 5 software. The primers are in the green box and the probes are in the red box.

**Figure 2 viruses-15-01789-f002:**
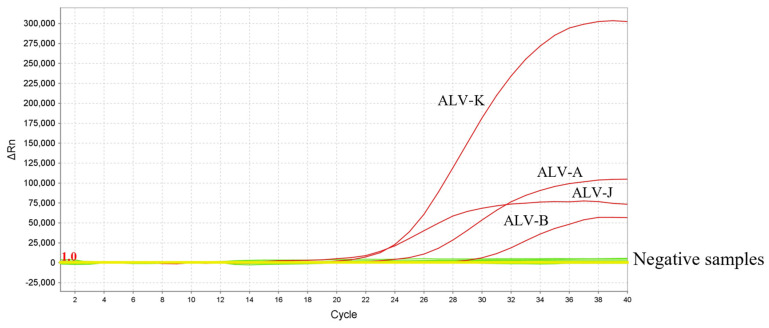
The specificity of the multiplex qPCR. Only ALV-A, B, J, and K subgroups strains produced four positive fluorescence signals, and no positive signals were observed with other avian pathogens (H9N2, NDV, IBV, REV, ILTV, *M. synoviae*, *M. gallisepticum*, MDV, FadV4, and ALV-E) and nuclease-free water.

**Figure 3 viruses-15-01789-f003:**
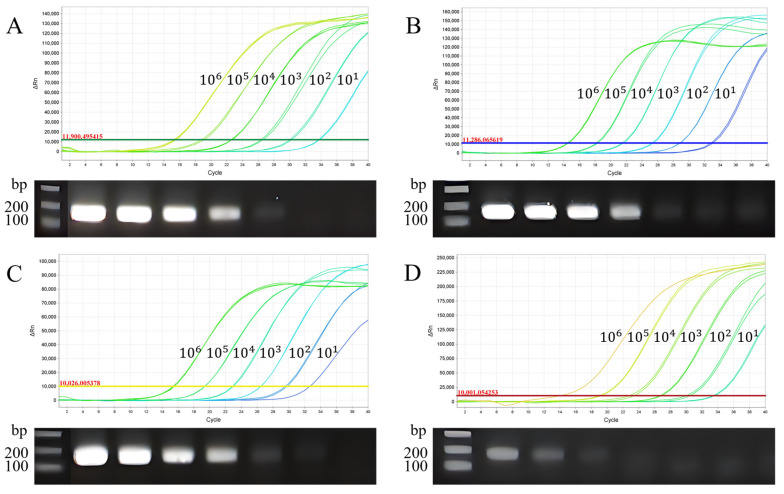
Sensitivity of the multiplex qPCR assay for detecting (**A**) ALV–A, (**B**) ALV–B, (**C**) ALV–J, and (**D**) ALV–K. Sensitivity tests were conducted using a 10-fold dilution series of recombinant plasmids, ranging from 10^1^ to 10^6^ copies/µL as templates, and the results were compared to the PCR results.

**Figure 4 viruses-15-01789-f004:**
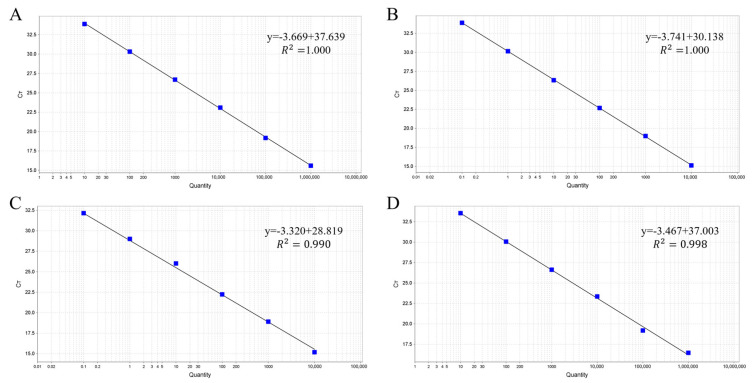
Standard curves of the multiple qPCR assay for detecting (**A**) ALV-A, (**B**) ALV-B, (**C**) ALV-J, and (**D**) ALV-K.

**Figure 5 viruses-15-01789-f005:**
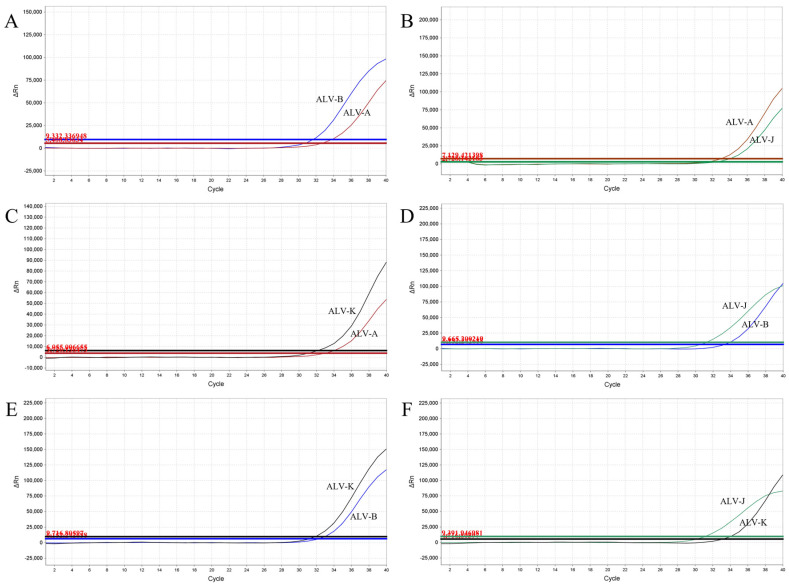
Coinfection simulation experiments with two ALV subgroups. Amplification curves of (**A**) ALV-A and ALV-B, (**B**) ALV-A and ALV-J, (**C**) ALV-A and ALV-K, (**D**) ALV-B and ALV-J, (**E**) ALV-B and ALV-K, (**F**) ALV-J and ALV-K at concentrations of the LOD.

**Figure 6 viruses-15-01789-f006:**
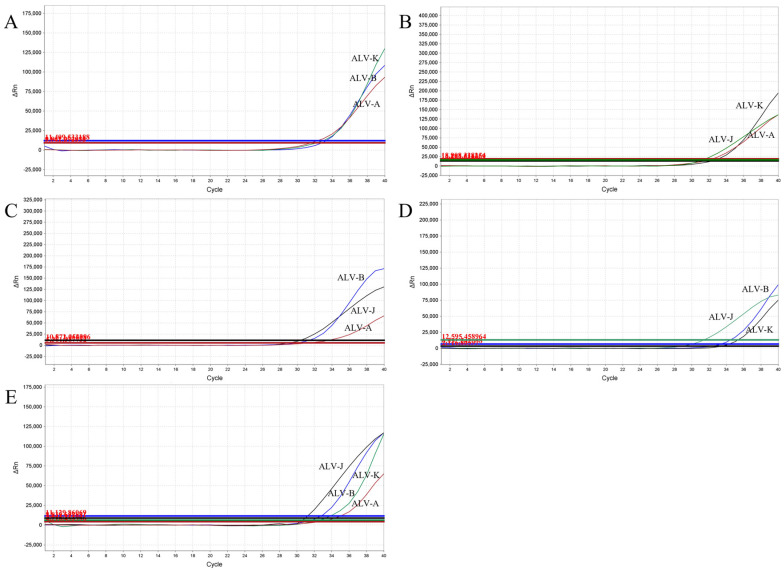
Coinfection simulation experiments with three and four ALV subgroups. Amplification curves of (**A**) ALV-A, ALV-B and ALV-K, (**B**) ALV-A, ALV-J and ALV-K, (**C**) ALV-A, ALV-B and ALV-J, (**D**) ALV-B, ALV-J, and ALV-K, (**E**) ALV-A, ALV-B, ALV-J and ALV-K at concentrations of the LOD.

**Table 1 viruses-15-01789-t001:** GenBank entries of the reference sequences.

ALV-A	ALV-B	ALV-J	ALV-K	ALV-E
KY612442.1	MG812183.1	MN735295.1	KM873219.1	KC610517.1
L10922.1	KC282901.1	MK683480.1	KP686144.1	AY013303.1
AF507031.1	JX570799.1	KF562374.1	MT783273.1	MF817822.1
MT179557.1	AF052428.1	MT538248.1	AB670312.1	KY235336.1
EU352877.1	MT648687.1	JN624880.1	KY490696.1	AY013304.1
MH186087.1	X00144.1	KU997685.1	KF999962.1	MF817821.1
KU375453.1	L10924.1	MW891540.1	MK941182.1	KC610516.1
KF866225.1	M11206.1	KY980662.1	KU605774.1	MT623675.1
HM452340.1	MT648688.1	JX423792.1	HM582658.1	EF467236.1
MZ836212.1	M14902.1	OL799232.1	KY767731.1	AB617818.1

**Table 2 viruses-15-01789-t002:** Primers and probes used in this study.

Primers and Probes	Sequences (5′ end to 3′ end)
qALV-A/K-F	TTCCCAGTCTCTCCCTAACATTACT
qALV-A/K-R	GCTGTCACCACCGTAAATGGTT
qALV-A-Probe	ROX-TTAGAAAAGGAGGATTGTYTAAGGAGGAA-BHQ2
qALV-K-Probe	FAM-AGTTGCGGCCTGGACCAATCTGAAA-BHQ1
qALV-B-F	CTACAGGTTCTGGGAAATGTACAAT
qALV-B-R	CTGTGCTTGTGCACCAATTTTC
qALV-B-Probe	Cy5-AAATGAGWCAGAATTGGTCCATCTGTCA-BHQ2
qALV-J-F	AAGAAAGACCCGGAGAAGACAC
qALV-J-R	CTTATTTGCCCAGGTGACCC
qALV-J-Probe	VIC-CACGTTTCCTGGTTGTTGCAACAGATG-BHQ1
ALV-A/B/J-F	GGATGAGGTGACTAAGAAAG
ALV-A-R	AGAGAAAGAGGGGTGTCTAAGGAGA
ALV-B-R	ATGGACCAATTCTGACTCATT
ALV-J-R	CGAACCAAAGGTAACACACG
ALV-K-F	TATGGATCCGATGTTCACTTACTCGAGC
ALV-K-R	AGAGTCGACGATGCTTCGTTTACGTCTTA

**Table 3 viruses-15-01789-t003:** Repeatability of the multiplex qPCR assay.

Plasmid	No. of DNA Copies	Intra-Assay		Inter-Assay	
		Ct (Mean ± SD)	CV (%)	Ct (Mean ± SD)	CV (%)
pMD-alvA	4 × $10^{3}$	26.54 ± 0.19	0.72	26.62 ± 0.11	0.41
	4 × $10^{2}$	30.03 ± 0.08	0.27	29.81 ± 0.04	0.13
	4 × $10^{1}$	33.85 ± 0.22	0.65	32.98 ± 0.36	1.09
pMD-alvB	1.1 × $10^{3}$	23.67 ± 0.13	0.55	25.78 ± 0.11	0.43
	1.1 × $10^{2}$	27.34 ± 0.20	0.73	29.18 ± 0.14	0.48
	1.1 × $10^{1}$	31.28 ± 0.26	0.83	31.56 ± 0.30	0.95
pMD-alvJ	1.37 × $10^{3}$	24.80 ± 0.04	0.17	25.55 ± 0.08	0.31
	1.37 × $10^{2}$	27.70 ± 0.08	0.29	28.16 ± 0.07	0.25
	1.37 × $10^{1}$	30.83 ± 0.07	0.23	30.60 ± 0.25	0.82
pMD-alvK	9.6 × $10^{3}$	24.81 ± 0.19	0.77	24.17 ± 0.08	0.33
	9.6 × $10^{2}$	27.52 ± 0.23	0.84	27.50 ± 0.12	0.44
	9.6 × $10^{1}$	31.73 ± 0.27	0.85	31.28 ± 0.18	0.58

**Table 4 viruses-15-01789-t004:** Clinical samples examined by multiplex qPCR, routine PCR, ELISA, and virus isolation.

Samples	Total	ELISA	Multiplex qPCR	Routine PCR	Virus Isolation
Tissue	132	26/132	25/132	25/132	21/132
Albumen	401	62/401	44/401	41/401	30/401
Plasma	104	32/104	30/104	28/104	27/104
Semen	215	41/215	37/215	36/215	32/215
Total	852	161/852	136/852	130/852	110/852
Positive rate	-	18.90%	15.96%	15.14%	12.91%

**Table 5 viruses-15-01789-t005:** Cell cultures with clinical samples examined by multiplex qPCR and routine PCR.

ALV Subgroup	Numbers (Percent)		
	Multiplex qPCR	PCR	ELISA ^a^
ALV-A alone	0 (0%)	0 (0%)	-
ALV-B alone	0 (0%)	0 (0%)	-
ALV-J alone	100 (11.74%)	97 (11.38%)	-
ALV-K alone	0 (0%)	0 (0%)	-
ALV-A/J	6 (0.70%)	6 (0.70%)	-
ALV-B/J	3 (0.35%)	3 (0.35%)	-
ALV-K/J	2 (0.23%)	2 (0.23%)	-
ALV-A/B/K	1 (0.12%)	1 (0.12%)	-
ALV-A/B/J	1 (0.12%)	1 (0.12%)	-
Total	113/852 (13.26%)	110/852 (12.91%)	110/852 (12.91%)

^a^ Positive antigen can only indicate ALV infection, but it cannot differentiate different ALV subgroup strains.

## Data Availability

All the data are presented in this study.

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
