# Peer review of "A Multiplex Quantitative Polymerase Chain Reaction for the Rapid Differential Detection of Subgroups A, B, J, and K Avian Leukosis Viruses"

_viruses, 2023, doi:10.3390/v15091789_

Round 1

Reviewer 1 Report (Previous Reviewer 2)

Dear authors,

As you are most probably aware population density, ie cities, farms, and total populations that can be infected by viruses are key factors for virus circulation, genetic variation and pseudotyping in viruses such as ALV's. 

Unfortunately these factors are not considered since the beginning,  nor are the poultry conditions (animal density, feeding, air conditioning).

When talking of the field, authors should provide the readers with the general organization of the farm

Minor corrections are required

Author Response

Reviewer 2 Report (Previous Reviewer 1)

No issues were detected in the ms. I recommend publication

Author Response

Reviewer 3 Report (New Reviewer)

The reviewed manuscript is dedicated to the design and validation of multiplex PCR assay detecting avian leucosis virus (AL), a pathogen causing various poultry diseases. Here, authors designed multiplex PCR test for AL detection, assessed its sensitivity and specificity. The presented results are timely and interesting for scientists, specializing on the field of molecular diagnostics. However, a number of issues needs to be addressed before publication.

Major issues:

1.      3.1. Establishment of a Novel Multiplex qPCR — authors are encouraged to add in this section or in Supplementary information about the assay optimization, including results of primers titration and elongation temperature gradient. Also, criteria of optimal conditions need to be stated.

2.      2.6. Sensitivity of the Multiplex qPCR — LoD calculations need to be clarified as actual LoDs seem to be different from plasmid standards concentrations.

3.      3.6. Evaluation Using the Clinical Samples — authors are encouraged to add more information about the validation on clinical samples, especially, about discrepancy between various assays, the frequency of each strain, and coinfected samples. Plausibly, higher positive rate of ELISA can also be the result of AVL-E presence, which need to be further investigated.

Minor issues:

1.      Although single infections with ALV-A, ALV-B and ALV-K were not found, coinfections with different subgroup strains were identified: 0.7% for ALV-A/J, 0.35% for ALV-B/J, 0.25% for ALV-J/K, and 0.12% for ALV-A/B/K and ALV-A/B/J — authors are encouraged to clarify this sentence as it can be confusing in the present state.

2.      Authors are encouraged to specify in the Introduction section why differential diagnosis of AVL strains is necessary.

3.      2.5. Specificity of the Multiplex qPCR — what RNA concentrations were used in these experiments?

4.      Table 5. Cell cultures with clinical samples examined by multiplex qPCR and routine PCR — the ELISA column is empty.

Round 2

Reviewer 3 Report (New Reviewer)

Many thanks to authors for their efforts in editing of the manuscript and detailed replies to all comments. However, several changes are still necessary before publication.

1.      Plausibly, tables with Cq and RFU values would be more readable than Supplementary figures with fluorescent curves.

2.      Indeed, ELISA and virus isolation do not allow to determine the virus type. However, these methods are still suitable for AVL testing. Therefore, it is necessary to compare ELISA and virus isolation with PCR-based approaches for readers to know “false-positive” and “false-negative” results of each method. If not in the main text, such table or tables can be provided in the Supplementary file.

3.      Cell cultures inoculated with clinical samples were tested for P27 antigen using a

3.commercial ELISA. However, positive antigen can only represent ALV infection, but

3.can not distinguish which subgroup of ALV strains is infected. Thus, the ELISA

3.column is empty — this explanation needs to be added in the manuscript, as the table of concern can be confusing for readers in her current form.

Author Response

This manuscript is a resubmission of an earlier submission. The following is a list of the peer review reports and author responses from that submission.

Round 1

Reviewer 1 Report

The ms. describes an important assay to detect simoultaneously the presence of several subtypes of ALV. However, there are several problematic aspects which were not referred in the ms.

1. The presence of endogenous ALV-E is prevalent, and can mislead the diagnosis, however, that aspect was not referrred. No specific primers were designed, and it is not clear how ALV-E was amplified. Whereas the presence of ALV-E should have been investigated in each sample, that was not performed. It is not correct that ALV-E can be detected only in egg white, as the authors mentioned in the Introduction.  

2. The authors did not described the process of sample cultivation in tissue cultures, which cells were used, C/E or C/O which differ in their capability to allow replication of ALV-E alongside with the other ALV subtypes.

3. What were the "samples" diagnosed, only tissue culture cultivated viruses, or in vivo organs, as mentioned in M&M lines 77-82? What was the similarity of results obtained from both samples sources? Were the same samples examined in both forms? The authors mention % of positives, but it is not clear what were the samples. Moreover, to analyse the samples taken directly from the organ, in vivo samples, it is very important, because further replication in tissues cultures introduce many mutations and genomic exclutions, during the RT process.

Reviewer 2 Report

See below my comment to the editor

In need of a complete description of the industrial poultry Set up

Also in need of an explanation for such a high Cot value

Languagei  editing required 

See comment above